# Dual-Arm Adversarial Robot Learning

**Elie Aljalbout**
Technical University of Munich
`elie.aljalbout@tum.de`

**Abstract:** Robot learning is a very promising topic for the future of automation and machine intelligence. Future robots should be able to autonomously acquire skills, learn to represent their environment, and interact with it. While these topics have been explored in simulation, real-world robot learning research seems to be still limited. This is due to the additional challenges encountered in the real-world, such as noisy sensors and actuators, safe exploration, non-stationary dynamics, autonomous environment resetting as well as the cost of running experiments for long periods of time. Unless we develop scalable solutions to these problems, learning complex tasks involving hand-eye coordination and rich contacts will remain an untouched vision that is only feasible in controlled lab environments. We propose dual-arm settings as platforms for robot learning. Such settings enable safe data collection for acquiring manipulation skills as well as training perception modules in a robot-supervised manner. They also ease the processes of resetting the environment. Furthermore, adversarial learning could potentially boost the generalization capability of robot learning methods by maximizing the exploration based on game-theoretic objectives while ensuring safety based on collaborative task spaces. In this paper, we will discuss the potential benefits of this setup as well as the challenges and research directions that can be pursued.

**Keywords:** Adversarial Learning, Robotic Manipulation, Learning Platforms

## 1 Introduction

With the recent progress in robot learning, many robotic tasks became possible to learn either via imitation or trial-and-error. These tasks range from classical robotics problems such as reaching [1, 2] and collision avoidance [3, 4] to more complex tasks involving locomotion [5, 6, 7] and manipulation [8, 9, 10, 11]. However, most methods that solve these tasks are either tested in simulation, assume to have access to a perfect estimate of the environment' state, or strongly restrict the interaction between the robot and its surrounding based on predefined heuristics or physical constraints. The main reason for these restrictions is the sample inefficiency of modern robot learning approaches, as well as the difficulty of safe environment exploration and resetting, without a human in the loop. Sample inefficiency is most commonly attributed to the complexity and noise encountered in sensory information processing. Solutions to the problem range from including pretrained perception modules [12] in the learning pipeline to integrating self-supervised state representation learning objectives into task learning [13, 10, 4, 14]. Safe exploration and environment resetting are rarely mentioned in publications and temporary solutions include engineering the environment or having a human manually stop the robot in dangerous situations and reset the environment at the end of each trial. While these approaches may work for simplified lab experiments, there is a clear need for scalable solutions in order to bring robots into real-world environments and human surroundings. In this paper we propose a simple paradigm to approach some of these problems. Namely, we propose dual-arm adversarial robot learning (DAARL) as a paradigm to automate robot learning experiments and ease the data collection. Besides the obvious benefits for dual-arm manipulation tasks, using a second hand makes it simpler to randomize single arm manipulation experiments, e.g. for the classical peg-in-hole task, a second robot can be used to hold the hole at different poses, allowing the learned policy to generalize to different settings. Additionally, for tasks requiring some sort of remote sensing such as vision, a second arm can be used to collect data for training perception modules. This data can either be used in a supervised fashion to train separate perception modules

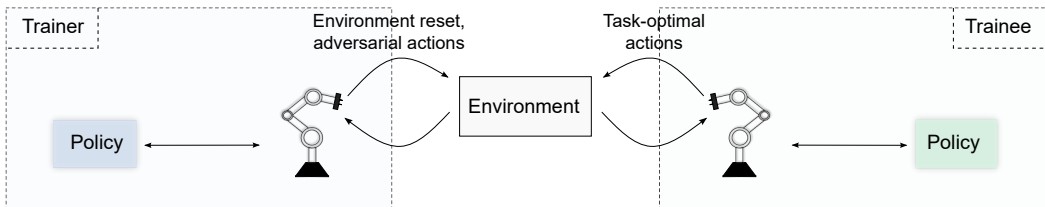

Figure 1: Illustration of DAARL: each robot is either a trainer or a trainee with its own policy. The trainee is the main learner. Its policy is trained to solve the task. The trainer policy and robot are responsible for resetting and randomizing the environment between episodes. It can also be used to generate adversarial actions to make the trainee policy more robust.

or even to integrate perception in the decision making (for instance by including state representation learning objectives in the task learning process). Furthermore, with the mutual awareness of the two robots, it is possible to design simple policies to solve the task at hand, and use these simplistic policies to sample data for training a more complex one. In this paper, we will introduce this paradigm (section 2), discuss its potential gains (section 3) and challenges (section 4), and discuss the general picture (section 5).

## 2   Methodology

Figure 1 illustrates the general idea behind DAARL. Each robot is either a trainer or a trainee. This condition can be relaxed for certain dual-arm manipulation tasks, but we would still use this naming convention for the sake of distinguishing the two robots. For single-arm tasks the trainer robot is only used to reset and randomize the environment for the task at hand, e.g. randomize the pose of the hole for single-arm peg-in-hole. The two robots are controlled by two separate policies, $\pi_{tr}$ controls the trainer and $\pi_{te}$ controls the trainee. The latter is the one performing the task while the trainer is the one responsible for resetting the environment and maximizing exploration. In addition, the trainer could also be used to generate disturbances during the task execution by the trainee. This behavior can also be adversarial similar to the work in [15]. It is important to note that the two policies don't have to run at the same frequency. For instance, if the trainer is only taking care of the environment reset of a single-arm task, it would only be used at the end of each episode, while the trainee policy is used at a higher frequency during the task execution. Considering a Markov Decision Process (MDP) as a tuple $\mathcal{M} = \{\mathcal{S}, \mathcal{A}_f, \mathcal{A}_l, T, \rho_0, r, \gamma\}$, where $\mathcal{S}$ denotes the state space, $\mathcal{A}_f$ and $\mathcal{A}_l$ the action spaces of the trainer and trainee policies. $T$ describes the system dynamics. $\rho_0$ defines the initial state distribution. $r : \mathcal{S} \times \mathcal{A}_f \times \mathcal{A}_l \to \mathbb{R}$ is the reward function , and $\gamma$ is a parameter for discounting future rewards. The overall objective for this problem can be formulated as follows:

$$\min_{\pi_{tr}} \max_{\pi_{te}} \mathbb{E}_{s_0 \sim \rho_0, a_{tr} \sim \pi_{tr}, a_{te} \sim \pi_{te}} \left[ \sum_{t=0}^{H} \gamma^t r(s, a_{tr}, a_{te}) \right] \tag{1}$$

As the solution to such problems can be complex [16], it might be desirable to train the two policies in an alternating fashion as described in [15]. This also makes it easier to control the degree of exploration by weighting the rewards for the trainer and trainee policies differently: $r_{tr} = \alpha r(s, a_{tr}, a_{te})$ and $r_{te} = \beta r(s, a_{tr}, a_{te})$. For dual-arm manipulation tasks, each robot would play the role of the trainer and the trainee, purely attempting to solve the tasks at times and tricking the other robot at other times. This would then require to have for each robot two policies, which are trained in an alternating fashion.

## 3   Potential Gains

**Safety during training.**  This aspect is especially relevant for single-arm tasks where the usual setup might include static (task-relevant) objects on surfaces (e.g. table) or objects in a human's hand. Due to the access to the trainer robot's state and hence the object's, it is possible to ensure safety by embedding this knowledge in the controllers and action spaces of both robots. Namely, it

is possible to predict and prevent robot-robot collisions, and subsequently avoid collisions altogether since these are mostly probable to occur around the object.

**Autonomous Randomized Experiments.** When learning via trial and error, it is important to be able to reset the environment, without having a human in the loop. For instance, if the task is "object pushing", it is necessary to reset the pose of the object after every training episode. While this is very simple in simulation, in real-world experiments, many problems can be faced such as objects out of reach, collisions, and having to switch/modify the end-effector to reset the environment. Having a second robot, makes it possible to fetch objects from a larger workspace and resetting them to an initial pose. Additionally, it is often desirable to randomize the initial state of the environment. Randomizing the initial state improves the generalization capability of the learned policy [17]. Besides reducing the dependence on a human to reset the environment, a second robot makes it possible to have more diverse initial states, since the objects to be manipulated can be placed in the trainer's hand which can be anywhere within this robot's workspace. Unlike the case where the object to be manipulated is placed in predefined positions on certain surfaces. Taking the example of a "pick and place" task, the trainer robot can hold the objects in hand (with the hand wide open) and the trainee attempts to pick the object. Of course, this has its limitations. For instance, the objects might fall outside of the workspace of both robots, a human is then required to return it into the scene. Finally, a dual-arm setup can also be beneficial for reward computation. For instance, if the object to be manipulated is rigid and always in the trainer's hand, it is possible to have access to its state without having to use any kind of external state estimation. This makes it easier to compute the reward.

**Human-Robot Interaction (HRI).** Another benefit of having a second robot, is to simulate a human for HRI tasks during training. Namely, the trainer robot can imitate the motion of a human that might be encountered during the task. By exposing the trainee to human-like motions during training, it is possible to train it for HRI tasks without a human in the loop. The trainer could also be designed to act in human-like ways. It is also possible to enforce certain properties on the trainer (e.g. time optimality) which are hard to impose on a human training the robot (for instance due to lack of motivation).

**Data Collection.** In Dual-arm settings, each robot can easily collect labeled data about the objects in the other robot's end effector, or ones previously moved by the other robot. The robots need to be calibrated with respect to each other and their kinematics should be known. For instance, for vision-based tasks, it is possible to collect datasets of images and object locations and use that later on for training segmentation, object detection, or tracking modules. This can also be done with a single-arm setting. However, the second arm makes the data collection faster, more flexible and enables collecting data about dynamic objects. Even for unsupervised/self-supervised state representation learning, collecting observations of the environment can be way better in the presence of an active agent (second robot) randomizing the state of the environment, and diversifying the observations. In both supervised and unsupervised cases, the obtained perception modules can later on be used for a variety of tasks even single-arm tasks. Another major benefit is to collect observation-action trajectories using simple (local) hand-designed teacher policies. The teacher policies could take advantage of the calibration between the two robots and generate actions based on the obtained knowledge concerning the accurate whereabouts of the task-relevant objects. Such policies can be hand-designed and sub-optimal. They can then be used to train a global policy as in [18]. More interestingly, the teacher policies could be fully blind (i.e. do not use vision and rely on true state information), but later on be used to collect data with images as part of the observation to train vision-based policies. Chen et al. [19] proposed a similar concept for training autonomous driving agents.

**Guided Exploration.** In robot learning, especially reinforcement learning-based methods, sample efficiency is a major problem. For that, a good balance between exploration and exploitation is very important to ensure generalization and escaping bad local optimas while also being sample efficient. However, both exploration and exploitation can be challenging. For exploration, a uniform random policy is already better than the task policy at collecting diverse samples. However, even a random policy can collect similar state-action trajectories, and hence more aggressive exploration is needed. The adversarial side of DAARL (section 2) help collecting samples where the current policy still struggles, instead of ones which are random but potentially very similar to the ones already encountered. Even for exploitation, most reinforcement learning methods need a lot of trials to learn a successful policy that can be exploited. As explained in the previous paragraph,

DAARL could accelerate that by simplifying the process of programming teacher policies based on the mutual awareness of the two robots.

# 4   Challenges & Limitations

Like any other paradigm, building DAARL-based pipelines and applications can be challenging. We discuss major challenges and limitations in this section:

**Increased Complexity:** First, by using a second robot for learning single-arm tasks, some extra engineering effort is needed. As previously discussed, the transformation between the two robots needs to be identified and a coordinated motion planning and control architecture is needed to ensure safety and smooth experiments[1]. In addition, every design and instrumentation choice needs to be made twice. For instance, the choice of action spaces and controllers used for every policy and robot are different. Additionally, for single-arm tasks, the trainer robot needs to be equipped with basic skills to be able to do its job on certain tasks. For example, if the role of the trainer is to reset and randomize the environment for a "pick and place" task involving multiple objects, it needs to be able to pick the objects itself and place it. At first, it might sound as if DAARL defies its purpose. However, the "pick and place" skill doesn't need to be perfect for the trainer but just good enough to serve its purpose, i.e. it could be hard-coded, sub-optimal, or based on object-specific assumptions. Nonetheless, that's an extra effort needed to get started. This motivates adopting a developmental approach to learning skills with DAARL: starting with basic atomic skills that require the least expert knowledge, and using the learned skills later-on while learning different skills and solving different tasks. Early tasks could even be supervised by a human.

**Extra Cost:** Besides the additional engineering efforts required to get started with DAARL, there's an extra cost needed to own the second robot, run it and maintain it. However, one of the main motivations for robot learning research is to enable robots to perform multiple tasks in unstructured and unknown environments such as households. We believe the actual robots deployed for such environments won't be single-arm robots attached to a static table, but rather something more dexterous and mobile like a humanoid. We might as well start training robots in these settings, solve some of the challenges that are faced in such settings, and take advantage of all the information that can be gained.

**Limitations:** As previously mentioned, DAARL doesn't completely remove the need for humans in the loop during training. We hope that it at least reduces the need for human supervision, and potentially eliminate it for certain tasks. Furthermore, the adversarial part of DAARL introduces additional hyperparameters and implementation details that need to be well chosen. For instance, in section 2, we suggested an alternating training procedure to solve the adversarial problem. Our choice was mostly based on our own previous experiences in similar optimization problems as well as related work [15]. Other alternatives might improve the training, reduce the number of hyperparameters and potentially strike a better balance between exploration and exploitation.

# 5   Discussion

In this paper, we propose dual-arm adversarial robot learning as a new paradigm for automating real-world robot learning experiments, reducing the dependence on humans in the loop, and boosting the exploration and generalization capabilities of such methods. Our main aim is to propose a feasible alternative to training in simulation, while reducing the chance of falling in common pitfalls of real-world robotic experiments, especially ones involving randomness. We also discuss other benefits of DAARL such as data collection, human-robot interaction, and safety-enabling potentials. We also discussed challenges and limitations of DAARL-like systems. Most importantly, we note that some of the majors advantages of DAARL, such as requiring less human interventions, are only limited to certain tasks. For that we briefly propose a developmental skill and task learning approach based on sorting the dependencies of tasks and skills on each others. Future work could attempt to smartly apply DAARL to various tasks, solve some of the mentioned challenges, improve the concept further, and even propose other smart mechanisms and platforms for real-world robot learning.

---

[1]These components are required anyway for dual-arm tasks

**Acknowledgments**

We would like to thank the reviewers for their valuable feedback and comments.

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
