# OpenReview forum: "Dual-Arm Adversarial Robot Learning"
_robot-learning.org/CoRL/2021/Conference/Blue_Sky — CoRL 2021, Blue Sky_

### Official Review · Reviewer_uiLR · 2021-08-26

**Novelty:** Good
**Impact:** 4
**Clarity Of Presentation:** Excellent

**Recommendation:**

Strong Accept: I recommend accepting the paper and will argue for my recommendation even if other reviewers hold a different opinion.

**Summary:**

Compared to a classical single arm setup the idea in the paper is to use a second arm in order to automate data collection and policy exploration in robot learning. A big challenge in robot learning is how to automatically gather high quality data. A second arm can function as part of the environment and thus change the environment to present a variety of training conditions to the first arm.

Most papers with real world robot experiments and robot learning structure the environment for the task to learn but here the idea is to have a more general approach that can be utilized for different learning tasks and not just a single one.

The paper discusses potential uses for the two arm setup such as improved safety, autonomous randomized experiments, human robot interaction, data collection, and guided exploration.

Using a second arm allows for collecting data about dynamic objects while having a setup that maintains safety. Robot policies can be optimized based on noisy sensor data while, assuming rigid objects and robot arms, safety can be guaranteed by tracking the object and arm locations. For example, instead of randomly throwing an object into the scene the second robot arm can move an object through the scene while maintaining accurate information about the positions of the robot arms and the object for safety.

**Summary Of Recommendation:**

The idea presented in the paper is interesting and can potentially lead to advances in the main challenge of robot learning, that is, how to collect good data efficiently and how to improve the process of robot learning.

The paper says that the second robot arm can be used to randomize the environment. For example, move a hole while the second robot tries to perform the peg-in-the-hole task. I like this idea and can see many uses for this. Changing the environment in different ways using a second arm makes sense. Many of the other ideas also make sense. I comment further on environment resetting below.


COMMENTS:

The idea of using the second robot arm to reset the environment seems to be limited to very specific situations. Typically resetting is needed when the first robot arm has either finished successfully or failed. Taking the peg-in-the-hole as a failure example, the first arm could have dropped the peg. But why is the second arm then needed? The first arm could just pick up the peg and move to the initial pose? This applies to most tasks? If the second arm is able to reset the environment, the first arm usually could have done the same thing without the second arm? The second arm is also not needed for moving objects on a table to specific places since the first arm can do that. I agree that the second arm can randomize the initial position of a potentially moving object that it grasps but I think for the general problem of resetting the environment the advantage of a second arm may be limited.

In
"Sample inefficiency is mostly attributed to the complexity and noise encountered in sensory information processing.",
what about, for example, complex discontinuous unknown dynamics? Why sensory information processing is the main challenge needs to be discussed in more detail.

What about triple-arm setups, or, n-arm setups? How do the ideas transfer to more than two arms? What about other kinds of actuators than arms? Or what about different end effectors such as suction based?

In the sentence
"This is due to the additional challenges encountered in the real-world, such as noisy sensors and actuators, safe exploration, non-stationary dynamics, autonomous environment resetting as well as the cost of running experiments for long periods of time.",
I assume "autonomous environment resetting" refers to how e.g. reinforcement learning agents are typically trained by resetting the environment to a specific state and this needs to be also done in robot learning which may be hard.
This may however not be at all clear to readers who are not familiar with robot learning and even when familiar with robot learning it is ambiguous. I suggest to phrase this part in a different way.

In
"there is a clear need for scalable solutions in order to bring robots into human environments",
"human environments" is mentioned the first time. I suggest to change "human" to describe the targeted environments better or introduce "human environments" earlier.

---

### Official Review · Reviewer_7mra · 2021-08-29

**Novelty:** Very Good
**Impact:** 3
**Clarity Of Presentation:** Good

**Recommendation:**

Weak Accept: I recommend accepting the paper, but will not argue for my recommendation if the majority of other reviewers have a different opinion.

**Summary:**

The paper aims to address the issue of robot autonomous learning with the specific scenario of dual-arm manipulation task, proposed a learning framework DAARL, and discussed its potential benefits and challenges.

**Summary Of Recommendation:**

The general idea of this research is really interesting. The main methodology is clearly described. And the analysis and the discussions of the proposed learning framework on its possible merits and demerits seem to be reasonable. I think this is a good Blue Sky paper.

---

### Official Review · Reviewer_mNei · 2021-08-30

**Novelty:** Good
**Impact:** 4
**Clarity Of Presentation:** Excellent

**Recommendation:**

Strong Reject: I recommend rejecting the paper and will argue for my recommendation even if other reviewers hold a different opinion.

**Summary:**

Proposes an interesting observation that learning with dual arm can be used as a stepping stone to scale robot learning in the real world. Authors make keen observations and sensible arguments about how dual arms setups can be used as a proxy to avoid the need for environment instrumentation for resets, state estimations etc. The HRI argument is a bit of a stretch but all other arguments around safety, randomization, curriculum, etc have some merit.


**Summary Of Recommendation:**

I actually like this idea but that is about it
- If this was a paper submission where authors demonstrate all they claim, I would have argued strongly for acceptance.
- The idea is nice but its not a unique idea. Many paper [1] [2] have previously leveraged this idea.
- The idea is just not out-there/ far-enough for me for it to be a "blue sky idea".

[1-webpage] https://sites.google.com/view/pddm

[1-video] https://youtu.be/nlJUOn3O1Ew

[2-webpage] https://sites.google.com/view/deeprl-dexterous-manipulation

[2-video] https://youtu.be/jxzoLDzcMeM

---

### Decision · Program_Chairs · 2021-10-01

**Decision:**

Accept

**Comment:**

The presented idea of using two arms for experimentation is simple and intuitive and good examples are given where the dual-arm approach is needed. While one reviewer argues that the paper is missing novelty, I think it still contains enough food for thought to create follow up papers and should be accepted.